# Data-Driven Low-Frequency Oscillation Event Detection Strategy for Railway Electrification Networks

**DOI:** 10.3390/s23010254

**Published:** 2022-12-26

**Authors:** David Gonzalez-Jimenez, Jon Del-Olmo, Javier Poza, Fernando Garramiola, Patxi Madina

**Affiliations:** Faculty of Engineering, Mondragon Unibertsitatea, 20500 Arrasate-Mondragon, Spain

**Keywords:** fault detection, low-frequency oscillation, railway catenary, CRISP-DM, data mining, machine learning, fault diagnosis, field dataset, data-driven

## Abstract

Low-frequency oscillations (LFO) occur in railway electrification systems due to the incorporation of new trains with switching converters. As a result, the increased harmonic content can cause catenary stability problems under certain conditions. Most of the research published on this topic to date is focused on modelling the event and analysing it using frequency spectrums. However, in recent years, due to the new technologies linked to Big Data (BD) and data mining (DM), a new opportunity to study and detect LFO events by means of machine-learning (ML) methods has emerged. Trains continuously collect data from the most important catenary variables, which offers new resources for analysing this type of event. Therefore, this article presents the design and implementation of a data-driven LFO event detection strategy for AC railway network scenarios. Compared to previous investigations, a new approach to analyse and detect LFO events, based on field data and ML, is presented. To obtain the most appropriate detection approach for the context of this application, on the one hand, this investigation includes a comparison of machine-learning algorithms (support vector machine, logistic regression, random forest, k-nearest neighbours, naïve Bayes) which have been trained with real field data. On the other hand, an analysis of key parameters and features to optimize event detection is also included. Thus, the most significant result of this work is the high metric values of the solution, reaching values above 97% in accuracy and 93% in F-1 score with the random forest algorithm. In addition, the applicability and training of data-driven methods with real field data are demonstrated. This automatic detection strategy can help with speeding up and improving LFO detection tasks that used to be performed manually. Finally, it is worth mentioning that this research has been structured based on the CRISP-DM methodology, established as the de facto approach for industrial DM projects.

## 1. Introduction

The integration of modern power electronics systems is causing power quality and stability problems in electric grids around different sectors. This problem has emerged, for example, in wind energy generation [1] and applications with modular multi-level converters for HVDC [2]. Moreover, the railway sector is another example where similar problems have arisen due to the incorporation of trains with new converters [3].

As the years have passed, the standard EN50388-2012 that railway manufacturers must comply with has become more and more demanding trying to avoid these AC catenary problems [4]. This standard defines different types of instability events based on the harmonic content of catenary currents and voltages. However, an exhaustive analysis of this phenomenon is difficult since the conditions under which it occurs are very specific and difficult to replicate in the laboratory. This is because the occurrence of the events depends on the characteristics of the catenary, on the number of trains connected simultaneously to the same catenary sector, as well as on the type of consumption. This is why it has been hard to propose methods that are able to detect catenary instability problems automatically.

However, for some years now, rolling stock has been equipped with remote monitoring systems that allow detailed analysis of systems operation almost in real time. Having acquired dynamic time series and environmental operational constraints (EOC), data from each train open up new possibilities for the detection and even diagnosis of catenary oscillation events. This context presents an opportunity for the development of data-driven strategies for automatic fault detection and diagnostics. Although data are available, Big Data and cloud-based monitoring platforms do not have enough acquisition frequency, for now, in order to analyse the harmonic content of catenary voltage and current as proposed by the standards. Therefore, another detection approach is needed, being that data-driven detection approaches based on machine learning are more suitable for the application. 

This scientific paper presents the design and implementation of an ML-based strategy for the detection of LFO events in AC railway network scenarios. In order to achieve the main objective and obtain the most suitable detection approach for the application under study, on the one hand, this investigation includes a comparison of machine-learning algorithms (support vector machine, logistic regression, random forest, k-nearest neighbours, and naïve Bayes) which have been trained with real field data, and on the other hand, an analysis of key parameters and features to optimize event detection. In summary, the scientific contributions of this research are the following:An automatic fault detection strategy based on ML is designed and implemented. Most of the studies carried out to date have focused only on the LFO event modelling and analysis of the possible causes [5,6,7,8,9,10,11,12]. The authors in [7] also analyse the influence that the characteristics of the traction units have on the occurrence of oscillations. This could lead to hardware and control software design improvements to mitigate these events.The dataset used for both training and testing of the algorithms is based on field data. That is, the industrial partner has detected and recorded manually different LFO events during the operation of its train fleet, which facilitates the development of supervised ML methodologies. This differs from the rest of the articles, as most of them are based on simulation or testbench data.Most DM scientific works follow the same steps to prepare data, train ML models, and evaluate their results. However, they do not follow any industry standard, which could make it difficult to deploy in an industrial application. Therefore, in this research, the DM project is structured based on the CRISP-DM methodology, established as the de facto approach for industrial DM projects [13,14,15]. Figure 1 shows the CRISP-DM methodology and its main steps.

The structure of the article follows the CRISP-DM cycle, as shown in Figure 1. Section 2 is focused on the theoretical analysis of the phenomena under investigation, as well as on the available raw dataset. This is equivalent to the Business Understanding and Data Understanding steps from the aforementioned methodology. Section 3 covers everything related to data preparation. Section 4 summarizes the different tasks regarding the modelling step, where the different ML algorithms are trained. Finally, Section 5 copes with the deployment, explaining the tools and platforms that have been used.

## 2. Power Quality and Stability Problems in Railway Applications

As it was mentioned before, catenary instability problems occur due to the incorporation of new trains with modern converters, since these are the largest source of harmonics in the entire system [16,17]. Concerning instabilities, there are three main phenomena that, due to the interaction between the traction power supply network and the trains, jeopardize the quality of supply and the stability of the catenary: the harmonic resonance, the harmonic instability, and the low-frequency oscillation (LFO) [6,18]. The characteristics of each type of event are presented in [18] and are summarized in Figure 2. Formation conditions and characteristics of each phenomenon are shown, as well as the influential factors at train and network levels.

This investigation studies the possibility of applying automatic detection approaches for LFO events, as these have been detected by the industrial partner in greater quantity among the dataset collected from its fleet. LFO is characterized by a low-frequency envelope of catenary voltage and current caused by inter-harmonic components at $f_{0}\pm f_{osc}$ around the fundamental frequency ($f_{0}$) [9]. As [18] proposes, LFO events are detected in two types of scenarios:1.In a railway system which has adopted the rotary frequency converter (RFC) as the power supply solution. The typical oscillatory frequency ($f_{osc}$) is approximately 10–30% of the respective power system’s fundamental frequency ($f_{0}$).2.In a railway system equipped with a static frequency converter (SFC) as a power supply solution and where several trains with four quadrant converters are in standby mode (auxiliary loads) located in the same railway depot, away from the traction substation (TSS). This leads to a 0.6–7 Hz LFO of the catenary voltage.

According to [19], the factors that contribute more to the appearance of this kind of event are:The number of vehicles and the load current in the same network sector.The contact line distance. The longer the contact line is, the larger the source impedance that increases the probability of instabilities will be.The line side traction converter control parameter tuning.

Figure 3 and Figure 4 show snapshots (in RMS values) of an LFO event detected by the industrial partner. In this case, the phenomena happened while the train was stopped (in a depot).

Most of the research published so far analyses this phenomenon from an analytical point of view (modelling the system) [7,8,10,11,12,20,21,22] or by analysing the harmonic content of field data [3,23,24]. EN50388-2012 also proposes the frequency as one of the features to study in order to distinguish between healthy and faulty behaviour. Therefore, once an event occurs, it is relatively easy to detect it by looking for the corresponding frequency components. This is a simple solution that does not require major technical developments. However, frequency analysis requires the analysed signals to have a sampling rate fast enough to distinguish faulty harmonics and avoid aliasing. Although nowadays railway traction control units (TCU) sample catenary voltage and current at high frequencies (e.g., 20 kHz) for drive control purposes, these measurements are usually down-sampled to send them to remote monitoring platforms. Unless there is a specific need, variables acquired at high frequency are not remotely available. As a result, the bottleneck in these cases is the computational and the data storage capacity that monitoring platforms have nowadays.

In this research, the data related to electrical variable measurements are down-sampled to several milliseconds. Moreover, instantaneous measurements are not available since sinusoidal signals were processed and stored as RMS values. Figure 5 shows a detailed view of catenary and DC-Link voltage in per unit with an LFO event.

In this context, it is considered that a data-driven detection strategy helps to identify this type of event, even with under-sampled data. Furthermore, the industrial partner has already identified 35 LFO events in the remote monitoring dataset. This is a good initial source of information to start the steps of designing a supervised ML strategy for automatic detection of LFO phenomena.

## 3. Data Preparation

### 3.1. Data Selection

In order to create a suitable database for training and testing ML algorithms, the most important variables are selected from all the available ones in the main data transmission bus of the train. So, as the main objective is to deal with LFO (an electrical phenomenon), and the first variable selection round is focused on the electric variables related to the drive of the train and its catenary. Specifically, the following variables are selected with the help of expert knowledge: catenary voltage, DC-bus voltage, catenary current, motor stator currents, estimated power, speed, and torque. 

In addition, as a way of putting the results into context, additional EOC variables such as GPS latitude and longitude are included. These variables are intended to improve the explainability of ML algorithms and to help with identifying problematic railway network sections and scenarios.

Finally, the correlation between all the variables is analysed. In this way, redundant variables are eliminated while maintaining the highest possible level of information in the dataset. Thus, if a correlation between two variables is higher than 90 %, only one of them is considered. After analysing the correlation percentages between all the variables, a total of 29 elements are selected as the useful dataset from an initial dataset of 103 variables selected using expert knowledge.

### 3.2. Data Cleaning

Data cleaning consisted of:Standardizing and adapting variable names.Filtering null values and replacing them using forward and backward filling techniques.Detecting and cleaning outliers.

In the case of outliers, apart from atypical values due to acquisition anomalies, pantograph detachments posed an obstacle when identifying catenary oscillations, because both events differ considerably from normal operation. In order to prevent false alarms and based on expert knowledge, it is determined that catenary voltages above and below 25% with respect to the nominal value are not necessary to detect LFO events, so they are classified as outliers.

### 3.3. Data Construction

Once the available dataset is cleaned of null values and redundancies, feature engineering is applied. The goal of this step is to prepare a training dataset that best fits the ML algorithms, as well as to improve the performance of the models. 

For manual feature extraction, time-domain, frequency-domain, and time/frequency-domain techniques can be used. In this study, considering the sampling rate of the variables, techniques including frequency analysis are discarded. Hence, the approach of statistical time-domain feature extraction is performed. For each variable, the acquired event length is split into windows of *N* samples. Then, these windows are summarized, computing the statistical characteristics shown in Table 1.

As a result, the number of samples for each variable in a log is reduced, and the number of features increased. That is why, for one event log, the initial raw data window of 29 variable (columns) and 937 samples (rows) is transformed into 237 features and 93 samples. Figure 6 shows the time series of a variable (catenary voltage) and some of the extracted statistical features (mean and variance).

Even if the dimension of the raw dataset is reduced considerably, feature selection techniques are applied to decrease the dimensionality even more. To do so, an analysis of variance algorithm (F-test ANOVA) is used to rank the independent features depending on their explanatory capacity with respect to the dependent variable (target). In other words, this method provides the *N* best characteristics that show the best separability between classes of the target variable, based on variance calculations [25,26]. In this case, the target variable is the label that defines if a specific sample is part of an LFO or not. At the end of this process, the dataset dimension is reduced from 237 features to 30, enabling more efficient model training and testing.

Finally, the variables are standardized using a standard scaler, matching them to a standard normal distribution (SND). All the variables are converted to have a mean value of zero and a standard deviation of one. The objective of this operation is to improve algorithm efficiency.

## 4. Model Training

### 4.1. Training Phase

Once the dataset is prepared, the machine-learning algorithms should be trained. In a first attempt, it is considered that the basic supervised learning algorithms should be able to detect LFO events with the available information. Therefore, the following algorithms are selected to make a comparison between them and select the most appropriate one for this application:Logistic regression (LR).Support vector machines (SVM).Random forest (RF).Naïve Bayes (NB).*k*-Nearest neighbours (*k*NN).

The next step is to split the whole pre-processed dataset into two parts: 70% for training and 30% for testing. Furthermore, when training and optimizing the selected algorithms, the *k*-fold cross-validation (CV) technique is applied. This technique consists of randomly dividing the training dataset into *k* subsets, of which *k*-1 groups is used to train the algorithm, and the remaining group is used for validation (see Figure 7). This iterative process is repeated *k* times using a different subset as validation in each iteration. Therefore, the process calculates *k* classification metrics whose average is used as the result in the training.

The main purpose of CV is to protect these kinds of algorithms against overfitting, which are very prone to it. A model with overfitting is highly tuned for the training dataset and is able to predict these data very accurately. However, it fails to predict new data.

In order to optimize the algorithms, hyperparameters are tuned using the Grid Search function from Python’s SKLearn package. This method together with the CV process returns the optimal set of hyperparameters with respect to the accuracy loss function.

### 4.2. Model Assessment

Once different algorithms are trained, accuracy and F-1 score are used for the assessment of the models [27]. Table 2 shows the results.

In all the cases, the results show that in terms of accuracy and F-1 score, algorithms are nearly above 90%. Although the algorithm with the best scores is RF, further analysis is performed.

As it was shown before, new features are extracted via the 10-sample time-sliding window method, and the most 30 influential features are selected by the ANOVA F-test. As these two factors are randomly selected, several tests are performed for training the algorithms with different combinations of sliding window length and feature quantity. Knowing that in the future this algorithm may be applied to an entire fleet with dozens of trains and that the dimension of the input dataset will be key, the objective of this analysis is to see the influence of these parameters in the performance of the algorithms. In particular, the window lengths of 5, 10, and 20 samples and datasets with 5, 15, and 30 features are tested (see Table 3). Figure 8 and Figure 9 show the results of each algorithm for different setups.

All the scores with the different pre-processing configurations improve 90% of accuracy. However, the setups where the window length for feature extraction is composed of 20 samples (Set 3, 6, and 9) are the worst setups. This is because by trying to compress the information of *N* points into just one, some of the information is lost unintentionally. Therefore, the higher the value of *N,* the less information is transmitted to the algorithm (more information is lost). Furthermore, five algorithms and pre-processing setups have achieved accuracy values around 97% and high values in F-1 score (RF with Set 1, Set 4, and Set 5, *k*NN Set 5 and SVM Set 8). It is worth mentioning that Set 4 with random forest uses half as many features as Set 1, obtaining very similar results. This may be a determinant algorithm and setup selection factor during the industrial deployment phase in the future.

## 5. Deployment

Although the results shown above are a proof of concept, the development is performed using tools used by the industry. Data storage, pre-processing, and modelling are performed using the Amazon Web Services platform. As an example, AWS S3 is used as the main data storage service. In addition, Cloud9 and Amazon SageMaker are used for data pre-processing and algorithm modelling and testing. Moreover, Figure 10 shows a schematic of the developed workflow in AWS. As a result, using those structures in the future for DM projects will reduce the whole detection strategy deployment time.

In addition, a final test is performed simulating the conditions that the application will encounter in the real operation. Therefore, instead of feeding the algorithms with data snapshots of limited duration (minutes), new time series with an approximate duration of eight hours are introduced in the detection workflow. Knowing that in a real environment the detection strategy will have to analyse massive data, the objective is to see its efficiency in a more real environment.

Figure 11 shows the pre-processing and classification times of the five different setups. In general, accuracy and F-1 score maintain the values shown in Table 2, but RF outperforms the rest of the algorithms in terms of pre-processing and prediction times. On average, it takes 94% less time than *k*NN or SVM algorithms.

Finally, it is worth mentioning that the manual method used previously by the industrial partner to detect LFO events visually is compared to the new proposed strategy. Considering that in the worst-case scenario the shortest LFO event found previously lasted less than 1 minute, the analyst should check one by one 1-minute windows in an 8-hour log and see whether there is an LFO or not. This means analysing 480 windows visually. With an average time of 20 seconds per window, the analysis takes 1 hour and 36 minutes. This is the case where the information is analysed blindly, without further details of date or location. In contrast, automatic detection with the implemented strategy takes approximately 10 minutes, both for the pre-processing and the detection. Moreover, in the 8-hour log test, the automatic strategy found 6% more events compared to the previous manual analysis.

## 6. Conclusions

This paper presents a data-driven detection strategy for low-frequency oscillation events in AC railway catenaries. 

In contrast to previous publications, the main contribution of this article is the development of the automatic detection strategy of the anomalous events based on machine-learning algorithms, which is not found in the scientific literature. In addition, another important contribution is the use of real field data for the training and testing of the ML algorithms. This work shows that data availability strengthens and supports maintenance of industrial applications. 

Moreover, the CRISP-DM methodology is used in the development of the proposed solution, which is nowadays considered as the de facto approach in industrial data-mining projects. This has helped to understand and pre-process the historical dataset and in turn to train and evaluate the implemented classification algorithms. 

Due to the monitoring design already implemented in the train fleet, the acquisition frequency of the variables is not fast enough to analyse LFO events by means of frequency analysis, as required by EN50388-2012. However, the large dataset available is optimized by extracting new statistical features in the time domain and filtering redundancies between them by applying feature selection techniques. This allows the use of machine-learning techniques. Specifically, five different supervised classification algorithms (LR, RF, SVM, kNN, and NB) are trained using the fivefold cross-validation approach. 

The results in terms of accuracy and F-1 score show that the most suitable classification algorithm is random forest, reaching 97.1% of accuracy and 93.9% of F-1 score. Moreover, it is the fastest classification strategy analysed, considering pre-processing and prediction times. 

The general conclusion of the research is that, although the current monitoring does not meet the requirements to analyse LFO failure according to the regulation, the acquisition of a large number of variables, data, and casuistry (Big Data) opens the opportunity to deploy ML models that are proven to be effective and efficient in this application.

## Figures and Tables

**Figure 1 sensors-23-00254-f001:**
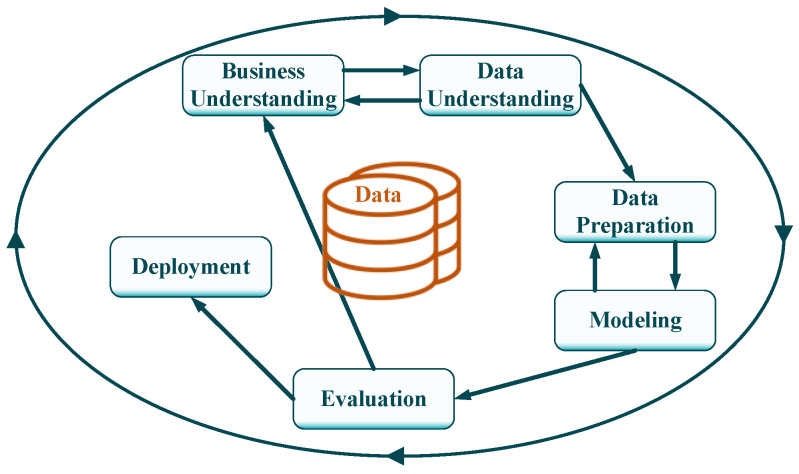
CRISP-DM methodology cycle.

**Figure 2 sensors-23-00254-f002:**
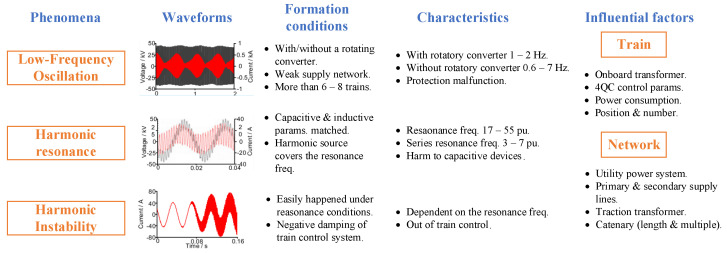
Main characteristics of the most common AC catenary instability phenomena: LFO, harmonic resonance, and harmonic instability [18].

**Figure 3 sensors-23-00254-f003:**
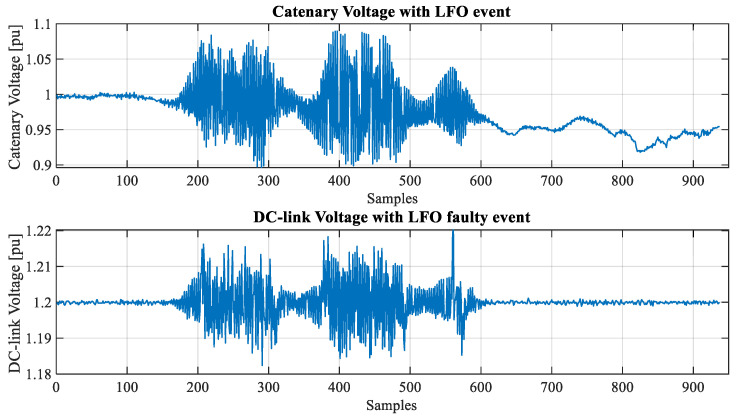
Catenary and DC-Link voltage (pu) under LFO faulty event.

**Figure 4 sensors-23-00254-f004:**
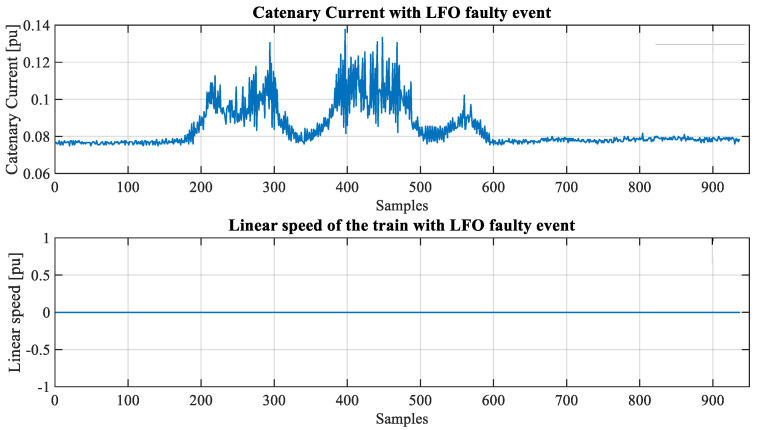
Catenary current and speed (pu) under LFO faulty event.

**Figure 5 sensors-23-00254-f005:**
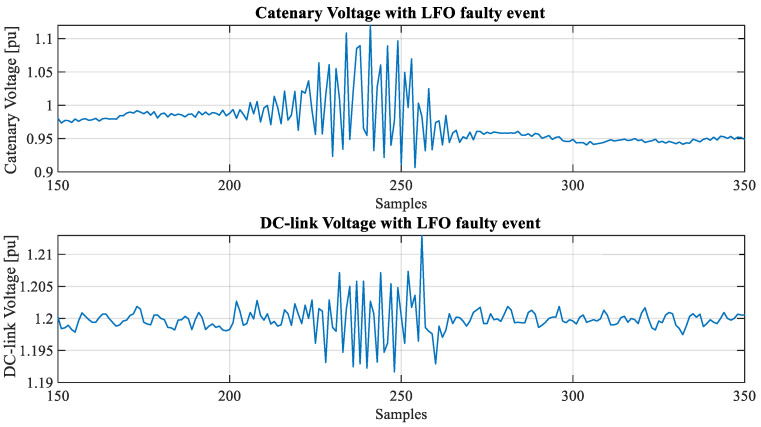
Detail of catenary and DC-Link voltage (pu).

**Figure 6 sensors-23-00254-f006:**
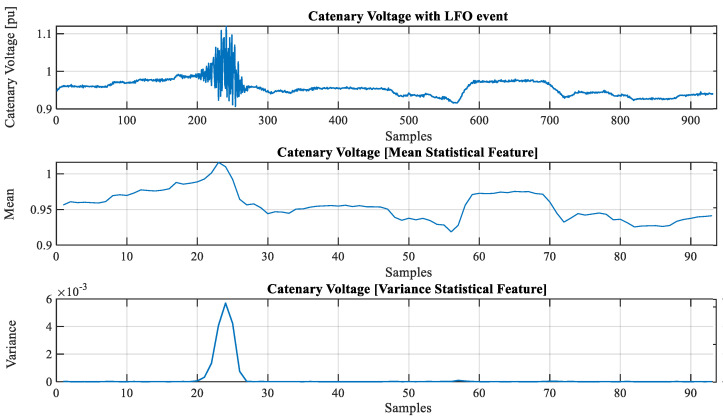
Statistical time-domain features from catenary voltage (mean and variance features).

**Figure 7 sensors-23-00254-f007:**
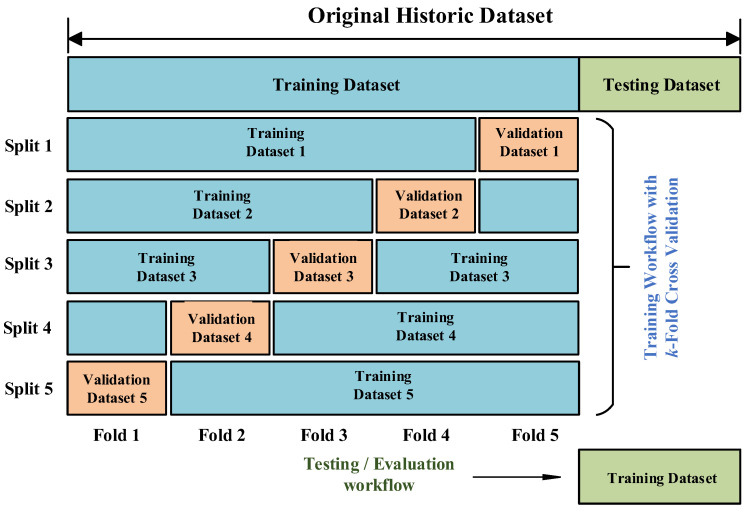
k-Fold cross-validation process.

**Figure 8 sensors-23-00254-f008:**
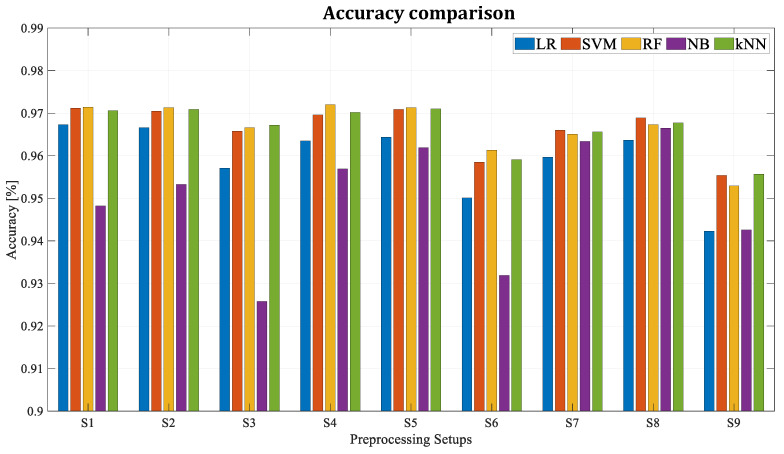
Accuracy for different algorithms and dataset configurations.

**Figure 9 sensors-23-00254-f009:**
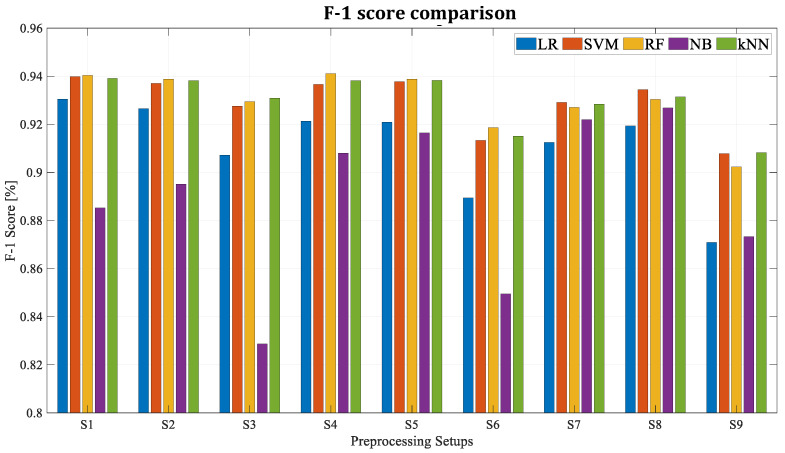
F-1 score for different algorithms and dataset configurations.

**Figure 10 sensors-23-00254-f010:**
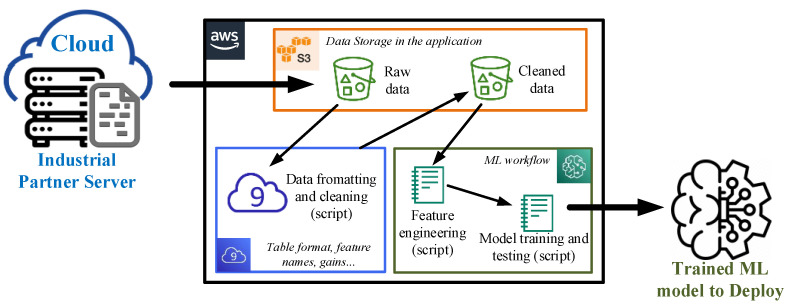
Data-mining architecture in Amazon Web Services platform.

**Figure 11 sensors-23-00254-f011:**
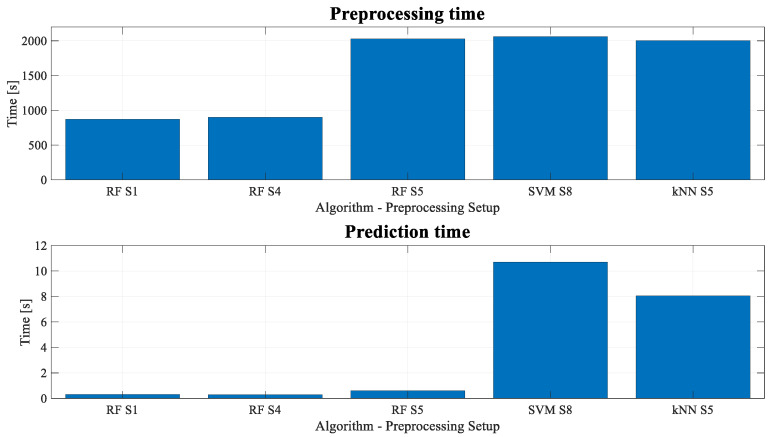
Pre-processing and classification times of the different setups.

**Table 1 sensors-23-00254-t001:** Time-domain statistical features extracted from the cleaned dataset.

Statistical Feature	Description
Maximum	$\mathrm{Max}\left(x_{i},\ldots x_{N}\right)$
Minimum	$\mathrm{Min}\left(x_{i},\ldots x_{N}\right)$
Average	$\frac{1}{N}\sum_{i=1}^{N}x_{i}$
Variance	$\frac{\sum_{i=1}^{N}{\left(x_{i}-\mu \right)}^{2}}{N-1}$
Standard Deviation	$\sqrt{\frac{\sum_{i=1}^{N}{\left(x_{i}-\mu \right)}^{2}}{N-1}}$
Range	$\mathrm{Max}\left(x_{i},\ldots x_{N}\right)-\mathrm{Min}\left(x_{i},\ldots x_{N}\right)$
Root Mean Square	$\sqrt{\frac{1}{N}\sum_{i=1}^{N}x_{i}{}^{2}}$
Kurtosis	$\frac{\sum_{i=1}^{N}{\left(x_{i}-\mu \right)}^{4}}{N\sigma^{4}}$
Skewness	$\frac{1}{N}\frac{\sum_{i=1}^{N}{\left(x_{i}-\mu \right)}^{3}}{\sigma^{3}}$

**Table 2 sensors-23-00254-t002:** Evaluation metrics of the algorithms after testing.

	LR	SVM	RF	NB	*k*NN
**Accuracy (%)**	96.7	97.1	97.1	95.3	97.1
**F-1 Score (%)**	92.7	93.7	93.9	89.5	93.8

**Table 3 sensors-23-00254-t003:** Dataset pre-processing setup description (S1–S9).

	S1	S2	S3	S4	S5	S6	S7	S8	S9
**Nº of samples/window**	10	5	20	10	5	20	10	5	20
**Nº of selected features**	30	30	30	15	15	15	5	5	5

## Data Availability

Not applicable.

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
