# Peer review of "Data-Driven Low-Frequency Oscillation Event Detection Strategy for Railway Electrification Networks"

_sensors, 2022, doi:10.3390/s23010254_

Round 1

Author Response

Dear reviewer,

Thank you very much for the effort taken to review this article, all your comments have be taken eninto account. I am attaching the document where you can find the letter that refers to each of your contributions. I hope that everything is understood correctly. The main changes are highlighted in yellow in the article.

Looking forward to hearing from you again,

David

Reviewer 2 Report

This paper proposes a data-driven low-frequency oscillation method for railway electrification networks. Previous studies focus on the analysis of the event based on the analytical modeling or spectral analysis. Owing to the development of the Big Data and Cloud Computing technologies, the massive data acquisition becomes possible. Therefore, the authors use the real data acquired through the machine learning method to analyze the LFO events. It sounds interesting, however, I have some concerns for this paper. The detailed comments are listed as follows:

1. This paper seems to be a report and it lack of the scientific nature. It is suggested that the authors to rewrite the abstract and the conclusion part. The abstract part should claim the experimental results. The conclusion part should be simplified.

2. I cannot understand the contribution of this paper. Except for the usage of real data, the main contribution of this paper should be emphasized using a separate paragraph in the introduction part. 3. In the experimental part, why not show the comparisons methods and performance?

4. In the references part, the cited references are out of date. The authors should cite more literatures from recent two years.

5. The experiments should be enriched because the current experimental results are sufficient to demonstrate the superiority of the proposed data-driven method.

Author Response

Dear reviewer,

Thank you very much for the effort taken to review this article, all your comments have been taken into account. I am attaching the document where you can find the letter that refers to each of your contributions. I hope that everything is understood correctly. The main changes have been highlighted in yellow in the article.

Looking forward to hearing from you again, 

David

Reviewer 3 Report

The authors of this paper performed the design and implementation of a Machine Learning procedure for the detection of Low Frequency Oscillation events in railway electrification due to switching converters. Real life data was used to train Machine Learning algorithms which is one of the strengths of this paper.  The authors have to address a small issue regarding the abstract. It should summarize the main work and findings. The abstract is somewhat lengthy and the authors have to include only their main contribution and results.

The topic is relevant as it introduces the use of artificial intelligence into the detection of low frequency oscillations.  

The key finding in this paper is the powerfulness of the Machine Learning algorithms in the detection of the LFO.

The authors have been successful in the adopted methodology. The use of raw filed collected data is meaningful as the events come from real life scenarios and not from fictitious ones through simulations. The training and testing of the machine learning algorithms using this dataset proved to be efficient and the results regarding the accuracy witness this.

The main conclusions drawn from this study are consistent with the objectives stated at the beginning of the paper. The machine learning algorithms improved the detection accuracy especially as it is based on real scenario datasets.

The list of references is very adequate and only relevant items are included.

Author Response

(The authors gave the same response as above.)

Round 2

Reviewer 2 Report

The authors have addressed part of my concerns, but I think there exists some deficiencies in the revised manuscript. For example, the abstract and conclusion should be reduced. The presented abstract and conclusion are too long.